# HPV Infection Significantly Accelerates Glycogen Metabolism in Cervical Cells with Large Nuclei: Raman Microscopic Study with Subcellular Resolution

**DOI:** 10.3390/ijms21082667

**Published:** 2020-04-11

**Authors:** Katarzyna Sitarz, Krzysztof Czamara, Joanna Bialecka, Malgorzata Klimek, Barbara Zawilinska, Slawa Szostek, Agnieszka Kaczor

**Affiliations:** 1Department of Virology, Chair of Microbiology, Faculty of Medicine, Jagiellonian University Medical College, 18 Czysta Street, 31-121 Krakow, Poland; katarzyna.sitarz@doctoral.uj.edu.pl (K.S.); barbara.zawilinska@uj.edu.pl (B.Z.); 2Faculty of Chemistry, Jagiellonian University, 2 Gronostajowa Street, 30-387 Krakow, Poland; 3Jagiellonian Centre for Experimental Therapeutics (JCET), Jagiellonian University, 14 Bobrzynskiego Street, 30-348 Krakow, Poland; krzysztof.czamara@uj.edu.pl; 4Centre of Microbiological Research and Autovaccines, 17 Slawkowska Street, 31-016 Krakow, Poland; joanna.bialecka@gmail.com; 5National Research Institute of Oncology, Krakow Branch, Clinic of Radiotherapy, 11 Garncarska Street, 31-115 Krakow, Poland; malgorzata.klimek@onkologia.krakow.pl

**Keywords:** human papillomavirus, glycogen, cervical cancer, cervical dysplasia, Raman microscopy, glycogenolysis

## Abstract

Using Raman microscopy, we investigated epithelial cervical cells collected from 96 women with squamous cell carcinoma (SCC) or belonging to groups I, IIa, IIID-1 and IIID-2 according to Munich III classification (IIID-1 and IIID-2 corresponding to Bethesda LSIL and HSIL groups, respectively). All women were tested for human papillomavirus (HPV) infection using PCR. Subcellular resolution of Raman microscopy enabled to understand phenotypic differences in a heterogeneous population of cervical cells in the following groups: I/HPV^−^, IIa/HPV^−^, IIa/HPV^−^, LSIL/HPV^−^, LSIL/HPV^+^, HSIL/HPV^−^, HSIL/HPV^+^ and cancer cells (SCC/HPV^+^). We showed for the first time that the glycogen content in the cytoplasm decreased with the nucleus size of cervical cells in all studied groups apart from the cancer group. For the subpopulation of large-nucleus cells HPV infection resulted in considerable glycogen depletion compared to HPV negative cells in IIa, LSIL (for both statistical significance, ca. 45%) and HSIL (trend, 37%) groups. We hypothesize that accelerated glycogenolysis in large-nucleus cells may be associated with the increased protein metabolism for HPV positive cells. Our work underlines unique capabilities of Raman microscopy in single cell studies and demonstrate potential of Raman-based methods in HPV diagnostics.

## 1. Introduction

Cervical cancer, associated with the sexually transmitted human papillomavirus (HPV) was the fourth most common cancer among women in the world in 2018 [1], although it is the most suitable cancer for both primary and secondary prevention [2]. This is mainly due to its slow development and wide access to screening and availability of vaccinations against HPV—the causative agent of almost all cervical cancers [2]. HPV is an 8000 base pair virus with eight early genes (*E1–E8*), two late genes (*L1*, *L2*) and long control region (LCR). Early genes code for non-structural proteins that regulate the maintenance and replication of the HPV genome in the infected cell, as well as the transition to the lytic cycle. The late genes in turn encode capsid proteins, which are also involved in the assembly of the virion [3]. Integration of the HPV genome in a host cell DNA is an important stage in the development of cervical neoplastic changes because it enables continuous overexpression of E6 and E7 oncoproteins [3,4]

Cytological examination, i.e., Pap smear, sometimes combined with an HPV test, is the basic method of cervical cancer screening [5]. However, a Pap smear is not a modern study—it was developed in the 1920s and is based solely on microscopic assessment that largely depends on the experience of a person evaluating the smear [6,7]. Additionally, the Pap test has a probability of a false positive cervical cancer result (0.8–1.2%) and, in particular, a false negative result that has been recently estimated to be 27.1% in the case of cervical cancer [8,9].

Tests for detecting HPV are divided into three groups: nucleic acid hybridization assays, signal amplification assays and nucleic acid amplification assays [10]. All tests detect DNA or mRNA of the virus. Tests can detect different types of virus, both low-oncogenic and high-oncogenic, and can specifically show infection with a specific type of virus or non-specifically indicate infection with one of the types of the group being detected [10].

According to the latest data, the HPV test is more effective at detecting cervical neoplasia vs. the Pap test. In fact, 2.3 and 5.5 of 1000 women developed CIN3 within 48 months of receiving a negative HPV and Pap test result, respectively [11]. However, the disadvantage of HPV testing is that the infection finally leads to neoplasia in only a very low percentage of women [12]. In addition, in some countries such as Poland, HPV infection testing is not a part of the national cervical cancer screening program [13]. Nevertheless, 99% of cases of cervical cancer coexist with infection of human papillomavirus [14]. HPV infection also causes most cases of oropharyngeal cancers and anal cancer [13]. Majority (ca. 95%) of cases of cervical cancer are associated with infection with high-risk HPV types, in particular strains 16 and 18 cause about 70% of cases of cervical cancer [15,16]. Interestingly, a viral infection can be cured spontaneously, especially for young women, which is why some researchers postulate not so much for a single detection of infection as for tests that confirm its persistence [17].

In the case of cervical cancer, the nucleus to cytoplasm ratio is used as a cytological feature to identify abnormal cells [18]. Interestingly, it does not depend on the amount of chromatin, which is also confirmed by research on yeast studies [18,19]. The size of the cell nucleus is primarily influenced by proteins associated with histone remodeling, such as histone acetyltransferase p300, internal nuclear membrane proteins, such as lamins and laminated proteins, and proteins related to the organization of the cell cytoskeleton [20,21,22]. In the first decade of the 21st century, when more and more works began to show that heterogeneity is a common phenomenon in cell populations, more attention was paid to this phenomenon in cell cultures [23]. The concept of “single cell analysis” was created in genetic and proteomic studies [24], but functioning of a cell cannot be fully understood solely based on its genome and proteome. In biochemical studies of cells, optical methods, particularly fluorescence microscopy, are becoming more and more popular. However, the use of labels may be associated with cytotoxicity and sometimes the procedure for its introduction into the cell is difficult and lengthy [25].

Therefore, other spectroscopic techniques are proposed, including Raman spectroscopy, a label-free, unbiased technique suited to study cells in their natural environment [26]. This method is based on the phenomenon of inelastic light scattering and is used to determine the chemical composition of samples [26]. A Raman spectrum is characteristic for a compound or a group of compounds enabling their identification in the tested sample. The combination of Raman spectroscopy and microscopy gives the opportunity to study non-invasively the distribution of chemical compounds with subcellular spatial resolution [27]. Moreover, with the incorporation of statistical methods, Raman microscopy becomes a powerful tool to study single cells and changes in them related to pathology development [28,29,30]. Some attempts have been made to identify neoplastic lesions using Raman spectroscopy [31,32]. Research on cervical cancer using Raman and infrared spectroscopy have been also carried out [33,34,35]. There has also been an attempt to detect HPV infections using Raman spectroscopy, where analysis of dry cell pellets was undertaken and global changes in the Raman signatures of cervical cells due to the presence of HPV infection were described [36]. Studies on cervical epithelial cells based on Fourier Transform Infrared Spectroscopy (FTIR) suggested that for dysplastic and neoplastic changes, a characteristic decrease in the intensity of glycogen-specific bands was observed [37]. Glycogen is one of the compounds accumulated intracellularly in animal cells in the form of stores with a diameter of 10 to 40 nm [38]. Already in the 1930s, Schiller determined that cervical cancer cells, unlike healthy cells, lack glycogen [39]. The most commonly accepted hypothesis explaining this phenomenon is the Warburg effect [40]. It assumes that cancer cells, often living under conditions with limited oxygen access, derive energy through glycolysis, not oxidative phosphorylation [40]. Additionally, recently, the Warburg reverse effect have been discovered, a phenomenon in which cancer-related fibroblasts (CAFs) provide cancer cells with substrates of the tricarboxylic cycle [41]. According to a recent study [42], this is not a one-sided CAFs action, but a two-sided interaction of CAFs and cancer cells. Cancer cells produce transforming growth factor beta 1 (TGF-β1), which by phosphorylation activates p38 mitogen-activated protein kinase (p38 MAPK kinase) in CAFs, which drives glycolysis in them. CAFs, in response to this stimulation, produce cytokines and chemokines that drive phosphoglyceromutase in cancer cells, which in turn leads to glycolysis and metastasis stimulation [42].

In the case of cervical epithelial cells, glycogen levels vary in different phases of the menstrual cycle, and after menopause it decreases significantly [43]. Interestingly, this phenomenon only affects ectocervical cells, while in endocervical cells, changes in glycogen levels during the menstrual cycle are small, and its level after menopause is not lower than on most days of the cycle in menstruating women [43]. In turn, in the case of cervical cancer, the level of glycogen decreases in both ecto- and endocervical cells, compared to the average level in the secretory phase [43].

Changes of the glycogen content in cancer cells are related to different signal transduction pathways in these cells [44]. Akt, i.e., protein kinase B is the main transmitter of the phosphoinositide 3-kinase (PI3K) pathway in cells [45]. A complex relationship has been found between Akt activity and the metabolism of carbohydrates in cancer cells [46]. In addition to Akt, glycolysis can also be promoted by p38 MAPK and Smad, all of these pathways show increased activity in cervical cancer and can be activated by TGF-β, which is a factor promoting metastasis [47,48]. Activation of the PI3K/Akt pathway is also associated with HPV infection. Both of its oncogenic E6 and E7 proteins affect the activation of Akt in infected cells [49,50].

It is apparent that changes in the cell metabolism in the course of tumorigenesis affect the chemical content of various cell components. As HPV induces activation of various signal transduction pathways related to metabolic activity of cervical cells, in this work, Raman microscopy with subcellular resolution has been used to study distribution of chemical components in epithelial cervical cells collected from HPV positive or negative patients. Patients were classified according to the Munich III system [51,52] to I, IIa, IIID-1 IIID-2 (IIID-1 and IIID-2 corresponding to Bethesda LSIL-low-grade squamous intraepithelial lesion and HSIL-high-grade squamous intraepithelial lesion groups, respectively) and cancer groups, in the followed text called I, IIa, LSIL and HSIL as Bethesda nomenclature is commonly used in the literature. Investigation of Raman data, supported by the chemometric data analysis and periodic acid-Schiff (PAS) staining, demonstrates that for cervical cells with large (diameter over 10 µm) cell nuclei, the level of glycogen is significantly decreased in comparison to low-diameter nuclei cells. Additionally, the glycogen level in both large and small-nucleus cells depends significantly on the dysplasia advancement and HPV presence showing clearly that metabolism of cervical cells is affected by HPV infection. This finding has a diagnostic potential and shows that transforming properties of HPV are related with accelerated glycogen metabolism.

## 2. Results and Discussion

### 2.1. Subcellular Distribution of Glycogen in Cervical Epithelial Cells

Raman imaging was used to analyze human epithelial cervical cells obtained from 96 patients classified to eight groups according to Munich III system [51] and HPV tests: I/HPV^−^, IIa/HPV^−^, IIa/HPV^+^, LSIL/HPV^−^, LSIL/HPV^+^, HSIL/HPV^−^, HSIL/HPV^+^ and cancer cells – squamous cell carcinoma (SCC/HPV^+^; for number of patients see Appendix A). The cluster analysis (CA, a statistical method enabling to classify spectra according to a chemical composition) was used to determine distribution of main subcellular structures. As the representative microphotographs (Figure 1A,C) and, in particular CA images (Figure 1B,D) show, the size of cell nuclei in epithelial cells varies significantly spanning the range of 4.5–15.8 µm (the class denoted in blue in Figure 1B,D, the marker Raman band due to the symmetric phosphodiester stretching vibration at 785 cm^−1^, Figure 1E).

Cytoplasm (denoted in orange) is rich in glycogen (Figure 1B,D) as the averaged spectrum clearly demonstrates due to presence of characteristic marker bands at 486, 577, 860, 941, 1083, 1129, 1335 and 1382 cm^−1^ (Figure 1E). Additionally, structures of changeable size and number spread in whole cytoplasm containing lipids (exhibiting characteristic bands due to lipids at 1307 and 1445 cm^−1^) and glycogen can be separated (denoted in maroon, Figure 1B,D). Glycogen plays an important role in early metastasis [42], fuels glycolysis in cancer cells [42] and it is known that its content decreases due to carcinogenesis [45]. Therefore, based on Raman microscopy and statistical methods, we evaluated the glycogen level in epithelial cells in all studied groups to determine how it is related to the nucleus size, cervical precancerous changes, and HPV infection.

### 2.2. Glycogen Content Decreases with the Increase of Nuclei Size of Cervical Epithelial Cells in I, IIa, LSIL and HSIL Groups, but Not the Cancer Group

In neoplastic cells, the diameter of nuclei is bigger than in healthy cells [53]. This phenomenon is used in pathomorphology to assess pathological changes in cells, including cervical cancer [53]. Subcellular resolution of Raman microscopy and the chemical characteristics of cell components enabled to separate cells differing by the size of nuclei. The chemical composition of nuclei for cells with nuclei of large and small diameter (defined arbitrarily as d ≥ 10 µm and < 10 µm, respectively) is, however, uniform independently on the group considered (Appendix A). It is in line with the finding that regardless of its size, the amount of chromatin in the nuclei of cervical epithelial cells is similar [18].

Nevertheless, the glycogen content (evaluated as the integral intensity of the characteristic band at 486 cm^−1^ both due to glycogen dispersed in the cytoplasm and glycogen in the form of glycogen-lipid-rich stores) occurs to be significantly decreased in cells with large cell nuclei in I/HPV^−^, IIa/HPV^+^ and LSIL/HPV^+^ (Figure 2). Additionally, in groups IIa/HPV^−^ and both HSIL groups there is a clear tendency to a reduced amount of glycogen in cells with large nuclei, although this relationship is not statistically significant, for HSIL groups quite obviously due to limited number of cells with the large cell nuclei. For the cancer groups the glycogen content is not related with the size of the nucleus.

A decreased level of glycogen in cervical cells is usually attributed to increased glycogenolysis [40]. Our results show that for cervical cells, particularly with cell nuclei of large diameter, show glycogen depletion that may be related with accelerated glycogenolysis. Moreover, the presence of HPV infection additionally influences the glycogen metabolism as the differences between the glycogen levels in cells with large and small-diameter cell nuclei are bigger in HPV^+^ groups compared to HPV^−^ (the decrease in the glycogen level for cells with nuclei d ≥ 10 compared to d < 10 µm equals for IIa/HPV^−^ vs. IIa/HPV^+^: 24% vs. 60%; LSIL/HPV^−^ vs. LSIL/HPV^+^: 5% vs. 60% and HSIL/HPV^−^ vs. HSIL/HPV^+^: 35% vs. 41%).

Patients classified as group IIa usually present as both cytologically and histologically negative. Interestingly, the presence of large-diameter nucleus cells of modified glycogen metabolism in the IIa/HPV^+^ group suggests that pathological changes already may occur in cells in this group, which until now were considered dysplasia-free, although this hypothesis undoubtedly requires further studies. To investigate in more detail how a glycogen level is influenced by the dysplasia progress and HPV presence, the statistical analysis of the cytoplasm glycogen content in cells from all groups was performed without separation for cells according to the nucleus size (Appendix A). Interestingly, our results show that if the large-diameter (that are less numerous in population of a given group) cell nuclei are not excluded, the cells in various groups behave as previously reported, i.e., the glycogen content is decreased in cancer cells [40], [54] and increased glycogen metabolism in HPV^+^ vs. HPV^−^ cells cannot be observed. It underlines importance of the single cells approach and subcellular resolution of Raman microscopy used in this work.

### 2.3. HPV Accelerates Glycogen Metabolism in Cervical Epithelial Cells

The results of the statistical analysis of the glycogen level (Kruskal–Wallis test, U’Mann–Whitney test) in all studied cells are presented in Figure 3 (also in Appendix A).

As we have clearly identified small-diameter nuclei cells as metabolically different that the large-diameter nuclei cells, we compared separately these two groups of cells. For cells with small nuclei (Figure 3A) for I/HPV^−^, IIa/HPV^−^, IIa/HPV^+^, LSIL/HPV^−^, and HSIL/HPV^−^ there are no statistically significant differences between the level of glycogen in the cytoplasm. The LSIL/HPV^+^ shows a difficult to rationalize increase in the glycogen level. Contrarily, the cells of patients with the cervical cancer have accelerated glycogen metabolism compared to all the above-mentioned groups, what agrees with previous findings showing that glycogenolysis is increased in cancer cells [40]. This effect is also observed by us for the for HSIL/HPV^+^ group confirming their phenotypic similarity to SCC/HPV^+^ cells.

In the case of cells possessing large-diameter cell nuclei (Figure 3B) the results are strikingly different. The level of glycogen in the cytoplasm of large-nucleus cells in HPV^−^ groups I, IIa, LSIL and HSIL is similar. However, the results clearly show a decrease of the glycogen content for cells infected with HPV that is statistically significant for IIa and LSIL groups and shows a trend for the HSIL group. Reduction in the glycogen level equals to 45% for IIa/HPV^+^ vs. IIa/HPV^−^, 46% for LSIL/HPV^+^ vs. LSIL/HPV^−^ and 37% for HSIL HPV^+^ vs. HSIL/HPV^−^, respectively. Moreover, these changes of the glycogen content are not dependent on age (Appendix A).

For the SCC/HPV^+^ group the average glycogen content does not significantly differ from most of the groups (Figure 3B). In combination with the data in Figure 3A, showing the lowest glycogen level for the SCC/HPV^+^ group, these results underline the importance of separate analysis of the cells with large and small cell nuclei. The key observation of this study is a very significant glycogen depletion in these large-nucleus cells in HPV^+^ groups compared with the respective HPV^−^ groups (results summarized in Figure 4).

Previously Curtis et al. [42] reported a considerable decrease in the glycogen level (the mobilization of glycogen stores) in late metastatic cells compared to early ones. The additional energy obtained in cancer cells is necessary to begin high-energy tasks such as migration, invasion, and further metastasis. It is hypothesized that the process of glycogen mobilization and possible increased glycolysis is used by large-nucleus HPV^+^ cells for increased protein synthesis [55]. We also conclude that the factors causing expanding of a cell nucleus considerably change also cellular carbohydrate metabolism and that it may be related with neoplastic transformation. This interesting phenomenological observation has to be studied in the future to understand a genetic background of this phenomenon. To confirm results obtained by Raman microscopy, the gold standard, i.e., PAS staining of the cells was performed (Figure 5).

The cells were divided into five groups, i.e., I, IIa, LSIL, HSIL and SCC (for number of patients see Appendix A). The additional separation due to the diameter of the cell nucleus was not possible due to the characteristics of PAS method (nuclear diameters could not be exactly determined). As Figure 5 shows, the average percentage of cells that were successfully stained with PAS is similar in groups I, IIa and LSIL, but significantly decreases with the progression of pathological changes, i.e., for HSIL and SCC groups. The results of PAS staining confirm the findings of spectroscopic measurements (Appendix A for direct comparison) validating the proposed Raman-based approach.

Our results show that a Raman-based methodology may be potentially used for fast HPV testing due to the decreased level of glycogen in the cytoplasm in HPV^+^ cells compared to HPV^−^ cells for a given group. Therefore, to preliminary evaluate possibilities of the Raman-based approach, the fiber probe Raman setup, applicable for application in healthcare facilities, was used to measure in a fast manner the glycogen content in a pellet of cervical cells showing that good quality Raman spectra of cervical cells can be obtained (Appendix A). Further large data study is necessary to confirm if infection of HPV is detectable using such an approach.

## 3. Materials and Methods

### 3.1. Clinical Specimens

In this study, samples from 96 women from the south of Poland and in the age of 19–85 years old were collected from October 2017 to February 2020 (Figure 6A, Appendix A). Cervical epithelial cells were obtained from The Centre of Microbiological Research and Autovaccines, in memory of Jan Bobr and from Department of Gynaecological Oncology, National Research Institute of Oncology, Krakow Branch. Research included women who underwent prophylactic check-ups (samples collected in The Centre of Microbiological Research and Autovaccines; samples from 77 women) and women during their first visit to the referral Oncology Center (samples from 19 women). Samples were taken prior to therapy and patients at this level did not undergo any therapeutic interventions, radiochemotherapy or biopsies before sampling. In the Oncology Center the samples were protected and stored until the results of colposcopy and histopathological examination were obtained. We included, in our experiments, cells of confirmed squamous cell cervical cancer.

Eight groups of patients have been analyzed in this study. The division into groups was based on the results of the cervical cytology test and classified according to Munich III and the PCR test for the presence of HPV infection. Research included patients whose epithelial cells were classified as group I, HPV negative; group IIa, HPV negative; group IIa, HPV positive; group IIID-1, HPV negative; group IIID-1, HPV positive; group IIID-2, HPV negative; group IIID-2, HPV positive (IIID-1 and IIID-2 corresponding to Bethesda LSIL–low-grade squamous intraepithelial lesion and HSIL–high-grade squamous intraepithelial lesion groups, respectively); and patients suffering from squamous cervical carcinoma (SCC), HPV positive. Due to only 1 and 2 samples in groups I, HPV positive and SCC, HPV negative, respectively, these groups were considered as statistically unreliable. Cells from every patient were divided into two parts and submitted for Raman imaging and PCR. For Raman imaging, freshly isolated cells were fixed using a 2.5% solution of glutaraldehyde for 4 min, then washed twice with PBS and stored in PBS in 4 °C until the measurement. Before Raman measurements, cells were placed on a Raman substrate (CaF_2_ slides, Crystran LTD., Poole, UK). For DNA isolation for PCR, cells were frozen in −20 °C until the assay was performed.

The trial was approved by the Bioethics Committee of the Jagiellonian University (23 Feb 2018, identification code: 1072.6120.29.2018). Written informed consents were obtained from all participants.

### 3.2. Raman Microscopy

Raman imaging was performed using confocal Raman spectrometer WITec Alpha 300 (Ulm, Germany) equipped with a UHTS 300 spectrograph (600 grooves·mm^−1^ grating, spectral resolution of 3 cm^−1^) and a CCD detector (DU401A-BV-352, Andor, Belfast, UK). A laser power of ca. 28 mW on a sample provided by a solid state 532 nm laser source was used. A 63× water immersive objective (Zeiss Fluor, NA = 1.0, Zeiss, Oberkochen, Germany) was applied to collect Raman spectra. Raman spectra were acquired with the integration time of 0.3 s for areas of 15 × 15 µm^2^ including cell nuclei with fragment of cytoplasm with the sampling density of 0.5 µm. For each patient at average 10 cells were measured (Figure 6B).

Pre-treatment of data—the procedure of cosmic rays removal and the background correction (polynomial of 3 degree) was carried out using the software WITec Project Plus 2.10. All imaging data were analyzed using the Cluster Analysis (CA, Figure 6C, K-means, Manhattan distance) to discriminate and separate the spectra from nuclei, cytoplasm and glycogen-lipid rich stores. The Opus 7.2 software was used for the vector normalization in the spectral range of 1500–400 cm^−1^, the averaging of the spectra in individual groups and calculations of the integral intensity of a band at 486 cm^−1^ due to the glycogen vibrations.

Statistical analysis of the data was performed using the Origin 9.1 software and the STATISTICA 13.3 software. The Shapiro–Wilk test was used to check whether the data met the assumption of normal distribution. Then, Kruskal–Wallis and Mann–Whitney U tests were performed (because data were not normally distributed). A Pearson’s correlation method was used to calculate the correlation between data. Two cells from groups IIa/HPV^+^ (big-diameter nuclei) and LSIL/HPV^+^ (big-diameter nuclei) and one cell from groups HSIL/HPV^+^ (big-diameter nuclei), LSIL/HPV^+^ (small-diameter nuclei) and SCC/HPV^+^ (small-diameter nuclei) was rejected from the analysis, as they were classified as outliers based on interquartile range.

### 3.3. PCR Reactions

PCR reactions for the presence of genital types of HPV infection were carried out using a nested PCR method and 2 pairs of primers: external MY09/MY11 and internal GP5+/GP6+ using Mastercycler Nexus × 2 from Eppendorf (Hamburg, Germany). To visualize nPCR effects, an agarose gel electrophoresis with addition of bromodeoxyuridine (BrDU) was performed. The test used detected fourteen types of highly oncogenic HPV (HPVhr): HPV16, 18, 31, 33, 35, 39, 45, 51, 52, 56, 58, 59, 66, and 68. For confirmation that only highly oncogenic types of HPV were analyzed, Cobas HPV (Roche, Basel, Switzerland) test was used.

### 3.4. PAS Staining

PAS staining was performed according to a known procedure [56]. Prior to staining, cells previously fixed with glutaraldehyde (2.5% solution, the same fixing procedure as described in Section 3.1) were centrifuged in a volume of 200 µL using a cytospin centrifuge (Shandon, Runcorn, UK; 200 rpm, 5 min) to fix the cells on a microscope slide. Cells were stained with 0.5% periodic acid solution for 5 min, placed in a Schiff reagent for 15 min and stained with Gill-modified hematoxylin for 90 s. Between staining, cells were washed with PBS buffer. The ratio of stained cells to total cells counted was under a light microscope (Axioskop, Zeiss, Oberkochen, Germany) at 10× magnification. Cells in two fields of view were counted for each sample.

## 4. Conclusions

In this study, we assessed differences in the cytoplasm glycogen level of cervical epithelial cells, collected from 96 women, depending on the presence of HPV infection, dysplastic changes and nucleus diameter using Raman microscopy and chemometric data analysis. The applied methodology appeared to be crucial to account for a considerable heterogeneity of cells in studied groups (I/HPV^−^, IIa/HPV^−^, IIa/HPV^+^, LSIL/HPV^−^, LSIL/HPV^+^, HSIL/HPV^−^, HSIL/HPV^+^ and SCC/HPV^+^). In particular, due to the subcellular resolution of Raman imaging, we were able to separate a subgroup of cervical cells with large (over 10 µm) diameter of nuclei, showing unexpected chemical and metabolic characteristics.

As our results demonstrate, in cervical cells glycogen is both dissolved in cytoplasm and aggregated with lipids in the form of glycogen-lipid-rich granules. For cells with small-diameter nuclei, the global level of glycogen in the cytoplasm is similar for considered groups apart from the SCC/HPV^+^ group that is characterized by a decrease in the glycogen content in agreement with previous studies [40] and HSIL/HPV^+^ group that is phenotypically similar to SCC/HPV^+^. Additionally, somewhat difficult to rationalize is an increase in the LSIL/HPV^+^ level. Importantly, there are no differences between the glycogen content for small-diameter cells in the HPV^+^ and HPV^−^ respective groups. Contrarily, for the subpopulation of large-nucleus cells, the cytoplasm glycogen level is significantly reduced of about 37–46% for HPV^+^ cells compared to HPV^−^ cells. It shows that for this subpopulation of cells, glycogen metabolism accelerates with HPV infection. The considerable depletion of the glycogen level in HPV infected cells may be associated with molecular pathways related with HPV E6 and E7 proteins [57,58] and increased energetic needs in the HPV infected cells for protein synthesis and virus replication. The mobilization of glycogen stores, i.e., increased glycogenolysis in late compared to early metastatic cells was previously attributed in ovarian cancer cells to their increased capabilities for migration and invasion [47]. Understanding of molecular basis of accelerated glycogenolysis in HPV infected cervical cells requires further research to shed light on mechanism of HPV-induced carcinogenic transformation.

Last, but not least, lack of reagents and speed of Raman spectroscopy could be advantageous in HPV diagnostics compared to the DNA-HPV method. Raman spectroscopy may be in future a base for a simple, automatized test for HPV; however, it certainly requires further testing on a bigger cohort of patients and improvement of the methodology.

## Figures and Tables

**Figure 1 ijms-21-02667-f001:**
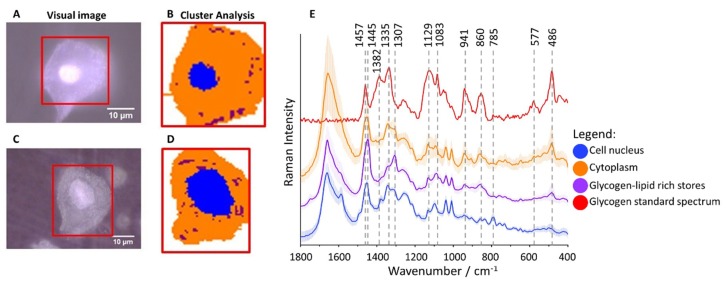
Glycogen-rich cytoplasm and stores in cervical epithelial cells. Microphotographs of representative cells ((**A**): IIa/HPV^−^; (**C**): IIa/HPV^+^) with a cell nucleus of diameter smaller and bigger than 10 µm, respectively, and CA images (**B**,**D**) showing distribution of cell nuclei (blue class) and glycogen/lipid-rich stores (maroon class) in the cytoplasm (orange class). The average Raman spectra of cell nuclei, cytoplasm and glycogen/lipid stores (**E**) averaged over all measured cells from all groups (in total 560 cells) along with the spectrum of the glycogen standard.

**Figure 2 ijms-21-02667-f002:**
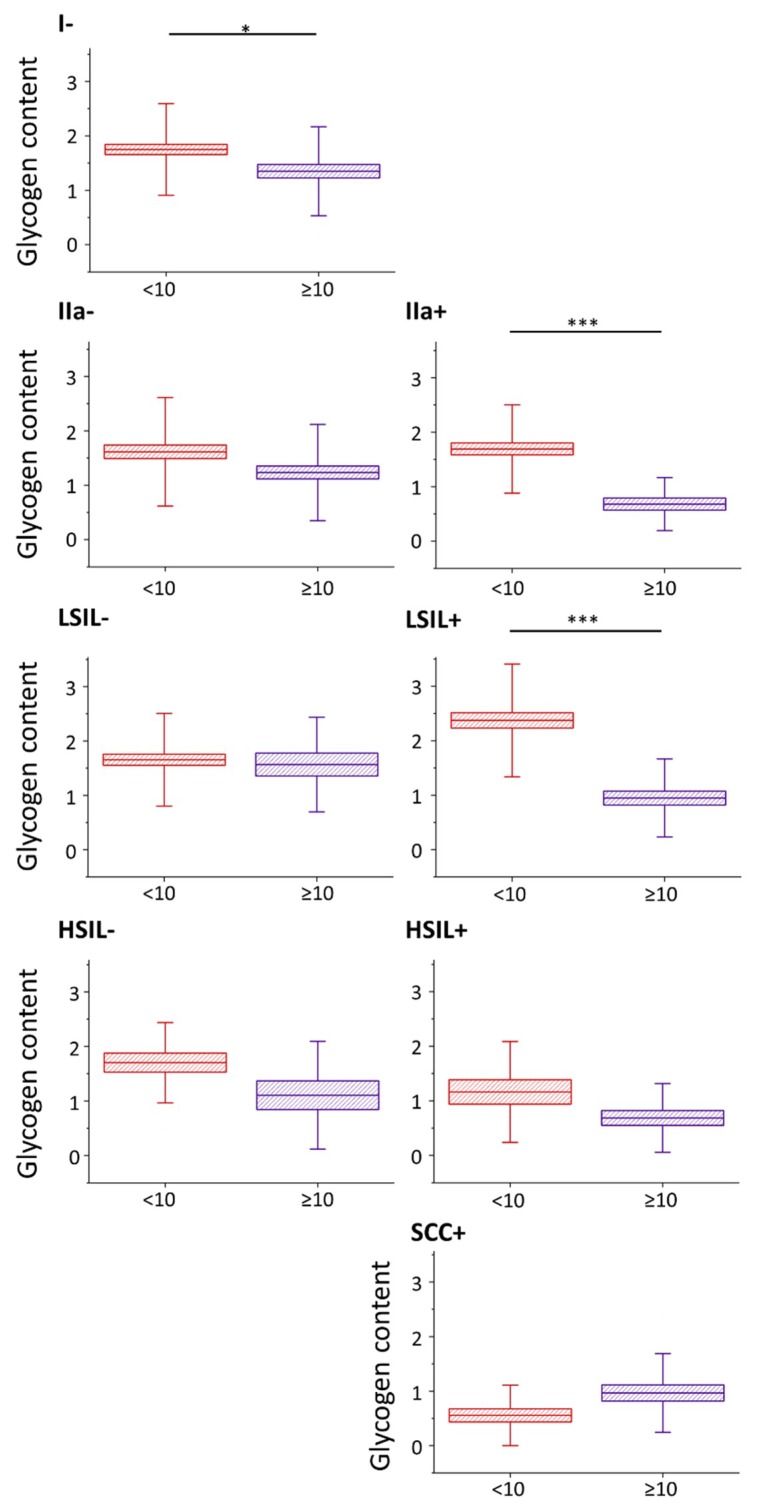
Glycogen content in epithelial cervical cells depending on the nucleus size. The analysis of the glycogen content in the cytoplasm of cervical epithelial cells with cell nuclei of a diameter <10 µm (burgundy color) and ≥10 µm (violet color) based on the integral intensity of the glycogen marker band at 486 cm^−1^. Graphs present individual groups of cells: I/HPV^−^, IIa/HPV^−^, IIa/HPV^+^, LSIL/HPV^−^, LSIL/HPV^+^, HSIL/HPV^−^, HSIL/HPV^+^ and SCC/HPV^+^. Mean values ± SEM are given as box plots: mean (horizontal line), SEM (box), SD (whiskers). * *p* < 0.05, *** *p* < 0.001.

**Figure 3 ijms-21-02667-f003:**
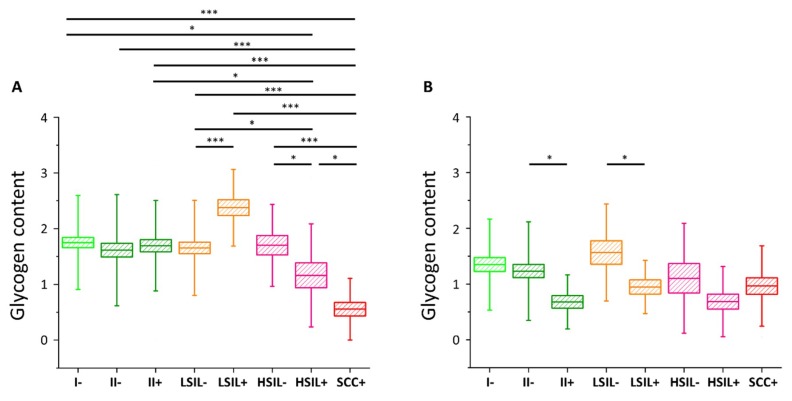
Glycogen content in the cytoplasm of cervical epithelial cells depending on HPV presence**.** The comparison of the glycogen content in the cytoplasm of cervical epithelial cells in studied groups: I/HPV^−^ (bright green), IIa/HPV^−^ (dark green), IIa/HPV^+^ (dark green), LSIL/HPV^−^ (orange), LSIL/HPV^+^ (orange), HSIL/HPV^−^ (pink), HSIL/HPV^+^ (pink), SCC/HPV^+^ (red) obtained by calculations of the integral intensity of the band at 486 cm^−1^ for cells with the nuclei of a diameter <10 µm (**A**) and ≥10 µm (**B**). Mean values ± SEM are given as box plots: mean (horizontal line), SEM (box), SD (whiskers). * *p* < 0.05, *** *p* < 0.001 (only key significances, described in the text, were presented).

**Figure 4 ijms-21-02667-f004:**
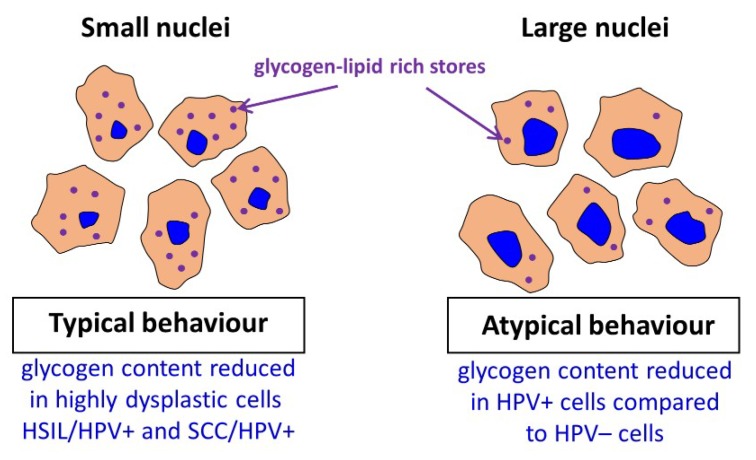
Summary of obtained key results.

**Figure 5 ijms-21-02667-f005:**
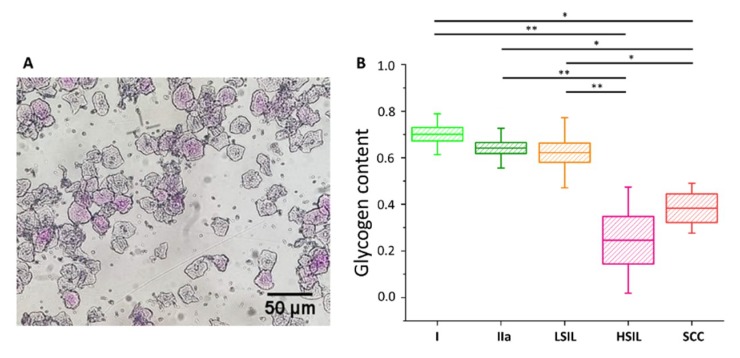
Glycogen content in the cytoplasm of cervical epithelial cells evaluated using a PAS staining method**.** An illustrative photograph of a sample of cervical epithelial cells stained by PAS (**A**). The analysis of the glycogen content (**B**) in the cytoplasm of cervical epithelial cells according to the division of patients into groups I (bright green), IIa (dark green), LSIL (orange), HSIL (pink) and SCC (red). Glycogen content was determined by optical evaluation in a light microscope after PAS staining by calculations of the average percentage of stained cells among all cells stained in individual groups. Each point on the graph corresponds to the average result for one patient (B). Values were given as mean ± SEM and were shown in box plots: mean (horizontal line), SEM (box), minimal and maximal values (whiskers). * *p* < 0.05, ** *p* < 0.01.

**Figure 6 ijms-21-02667-f006:**
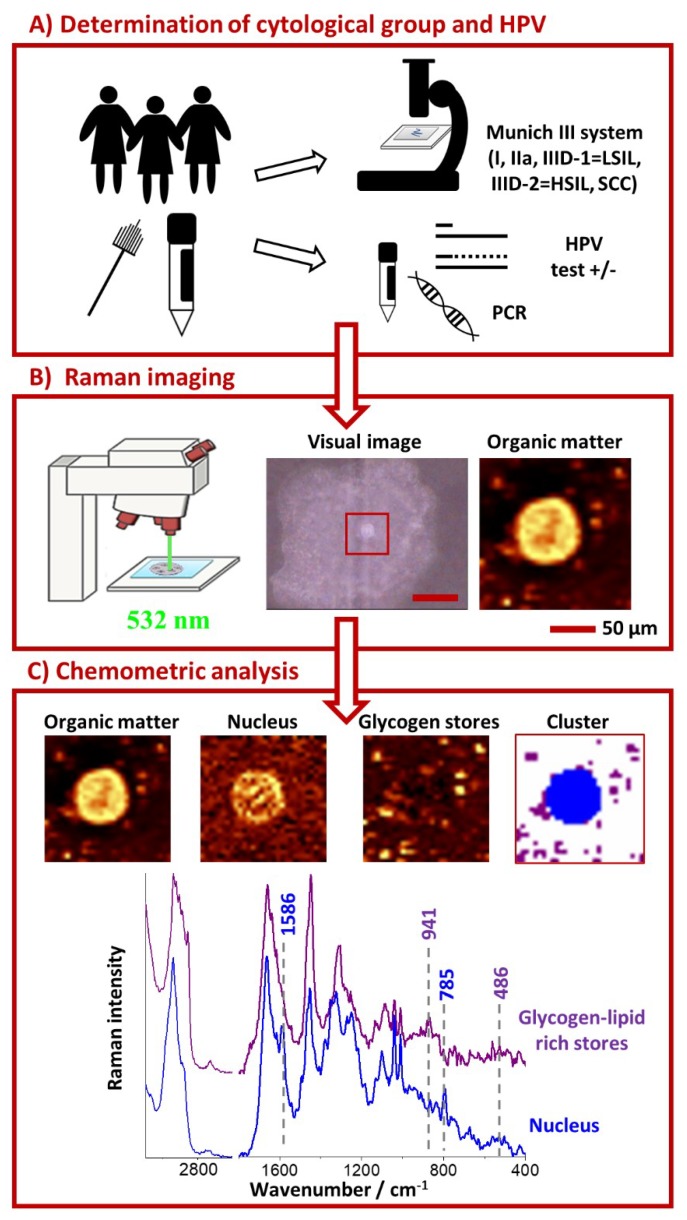
The scheme of the experiment layout**.** Cell cytological classification and testing for HPV infection (**A**), cell measurement by Raman microscopy (**B**) followed by Cluster Analysis and comparison of spectra between groups (**C**).

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
