# Peer review of "HPV Infection Significantly Accelerates Glycogen Metabolism in Cervical Cells with Large Nuclei: Raman Microscopic Study with Subcellular Resolution"

_ijms, 2020, doi:10.3390/ijms21082667_

Round 1
Reviewer 1 Report
Appreciate responses to comments.
Suggest that you specify the numbers ascertained from the two sources in the manuscript (as you did in response to reviewer)
Author Response
Appreciate responses to comments.
Suggest that you specify the numbers ascertained from the two sources in the manuscript (as you did in response to reviewer).
Response: We have specified in the manuscript the number of women ascertained from the two sources as suggested by the reviewer.
In page 9 the manuscript text is changed as follows:
"Cervical epithelial cells were obtained from The Centre of Microbiological Research and Autovaccines, in memory of Jan Bobr and from Department of Gynaecological Oncology, Maria Sklodowska-Curie Memorial Cancer Centre and Institute of Oncology. Research included women who underwent prophylactic check ups (samples collected in The Centre of Microbiological Research and Autovaccines; samples from 77 women) and women during their first visit to the referral Oncology Center (samples from 19 women)."
Reviewer 2 Report
ID IJMS-764860
"Impact of HPV infection on the cellular glycogen level: a new molecular mechanism in progression of cervical carcinogenesis. RAMAN microscopic study with subcellular resolution" to be published in International Journal of Molecular Sciences.
General remarks
The authors corrected their manuscript according to all comments.
Author Response
We want to thank the reviewer for approving our changes.
Reviewer 3 Report
The key observation of this study is the significant glycogen depletion in large-nucleus cells in the HPV-positive group compared to the HPV-negative group (lines 244 – 246). This is the case for the Pap group II and LSIL (low grade dysplasia) but not for HSIL (high grade dysplasia). It is not clear to the reviewer why the authors interpret these phenotypic differences as “ a new molecular mechanism in the progression of cervical carcinogenesis” as suggested in the title of the manuscript. After all, women with HPV-positive PapII and HPV-positive LSIL have only a very low risk for progression. Moreover, there are no significant differences in the glycogen content between HPV-positive Pap II, HPV-positive LSIL and HPV-positive HSIL (Figure 3B).
Another issue is the authors´ statement in the abstract (lines 30/31): “Our work underlines unique capabilities of Raman microscopy and demonstrate potential of Raman-based methods in HPV diagnostics”. The reviewer can appreciate the unique capabilities of the Raman approach concerning single cell analyses, it is however not clear how Raman-based methods are thought to be implemented in cervical cancer screening: Should Raman-based methods be used instead of HPV-DNA detection? Should the method be used instead of cytology, or for triage, or for risk assessment? This needs to be discussed in the paper in view of the date presented.
Author Response
The key observation of this study is the significant glycogen depletion in large-nucleus cells in the HPV-positive group compared to the HPV-negative group (lines 244 – 246). This is the case for the Pap group II and LSIL (low grade dysplasia) but not for HSIL (high grade dysplasia). It is not clear to the reviewer why the authors interpret these phenotypic differences as “ a new molecular mechanism in the progression of cervical carcinogenesis” as suggested in the title of the manuscript. After all, women with HPV-positive Pap II and HPV-positive LSIL have only a very low risk for progression. Moreover, there are no significant differences in the glycogen content between HPV-positive Pap II, HPV-positive LSIL and HPV-positive HSIL (Figure 3B).
Response: The reviewer is correct that “a new molecular mechanism“ may appear too far-reaching. Therefore we propose a new title, that hopefully better reflects the key points of the manuscript: “HPV infection significantly accelerates glycogen metabolism in cervical cells with increased nuclei: Raman microscopic study with subcellular resolution”.
Another issue is the authors´ statement in the abstract (lines 30/31): “Our work underlines unique capabilities of Raman microscopy and demonstrate potential of Raman-based methods in HPV diagnostics”. The reviewer can appreciate the unique capabilities of the Raman approach concerning single cell analyses, it is however not clear how Raman-based methods are thought to be implemented in cervical cancer screening: Should Raman-based methods be used instead of HPV-DNA detection? Should the method be used instead of cytology, or for triage, or for risk assessment? This needs to be discussed in the paper in view of the date presented.
Response: We believe that the advantage of Raman spectroscopy in HPV diagnostics compared to the DNA-HPV method is its speed and lack of reagents. We hope that Raman spectroscopy could be a base for a simple, automatized test for HPV, however it requires further research on a bigger cohort of patients and improvement of the methodology. In Poland, where the DNA-HPV test it is still quite expensive, the Raman-based test could enable testing more women.
We have added short discussion on this topic in page 13:
“Last, but not least, lack of reagents and speed of Raman spectroscopy could be advantageous in HPV diagnostics compared to the DNA-HPV method. Raman spectroscopy may be in future a base for a simple, automatized test for HPV, however it certainly requires further testing on a bigger cohort of patients and improvement of the methodology.”
Round 2
Reviewer 3 Report
The altered title of the manuscript is now in line with the data presented. To avoid confusion, the term "increased nuclei" in the new title should be replaced by "large nuclei".
Author Response
The altered title of the manuscript is now in line with the data presented. To avoid confusion, the term "increased nuclei" in the new title should be replaced by "large nuclei".
Thank you for this remark. We have changed the title accordingly.
This manuscript is a resubmission of an earlier submission. The following is a list of the peer review reports and author responses from that submission.
Round 1
Reviewer 1 Report
This study considers the association between combination of HPV infection and grade of cervical cytological abnormality and cellular glycogen level.It is a cross-sectional analysis in which techniques enabling the sub cellular distribution of glycogen in cervical epithelial cells to be characterized.
Comments
It is stated that samples of cervical cells were obtained from 87 women from southern Poland aged 19-76 as part of LBC testing. However, the women were ascertained through a microbiological research centre and an oncology department. Therefore, it is unclear to me whether these women had a clinical history of infection or cancer that required treatment/follow-up at tertiary referral centres. I note that there is an organized cervical screening program in which women aged 25-59 are invited to undergo cervical screening, with an uptake rate of less than 30% (see https://www.ncbi.nlm.nih.gov/pmc/articles/PMC4417537/). The clinical profile of women ascertained through the organized screening program would be very different from that of women ascertained from specialist tertiary referral centres The women could theoretically have been classified into eight groups, but two were excluded because numbers were very small. It would be helpful to specify the number of women per group. Its not clear to me what system of classification of cytological abnormality was used. I,II or III could referral's to the original classification of Papanicolaou of 1963, or one of the Munich systems (see https://www.ncbi.nlm.nih.gov/pmc/articles/PMC4664210/#b2-jtgga-16-4-203). The organized system in Poland is stated to use a modification of the Bethesda system (PMC4417537, as already cited). It is stated that the cells obtained by LBC were divided into two parts (one for Raman imaging, the other for HPV testing). How was this done? Could the method of division affect the relationship between imaging and HPV? LBC cytology does not necessarily map to results of histological examination of tissue samples. Referring back to point 1, had any or indeed all of these women undergone colposcopy with large loop excision of the transformation zone (LLETZ, also known as LEEP) or punch biopsy, or other intervention. Such procedures could affect presence of HPV infection and cytology. Indeed, for a mechanistic study, why use cytology rather than histopathological samples?
Author Response
Reviewer 1
Comments and Suggestions for Authors
This study considers the association between combination of HPV infection and grade of cervical cytological abnormality and cellular glycogen level. It is a cross-sectional analysis in which techniques enabling the sub cellular distribution of glycogen in cervical epithelial cells to be characterized.
We are grateful for all the comments that enabled us to improve our manuscript. We answered below and modified the text accordingly to the reviewer’s comment.
Comments
It is stated that samples of cervical cells were obtained from 87 women from southern Poland aged 19-76 as part of LBC testing. However, the women were ascertained through a microbiological research centre and an oncology department. Therefore, it is unclear to me whether these women had a clinical history of infection or cancer that required treatment/follow-up at tertiary referral centres. I note that there is an organized cervical screening program in which women aged 25-59 are invited to undergo cervical screening, with an uptake rate of less than 30% (see https://www.ncbi.nlm.nih.gov/pmc/articles/PMC4417537/). The clinical profile of women ascertained through the organized screening program would be very different from that of women ascertained from specialist tertiary referral centres The women could theoretically have been classified into eight groups, but two were excluded because numbers were very small. It would be helpful to specify the number of women per group.
Squamous cell cervical cancer was diagnosed in 11 women in stages according to FIGO classification IIA-IIIB. All cervical cells samples were obtained before radiochemotherapy.
The remaining women were tested for HPV infection as an additional test to the routine Pap smear. In many cytological laboratories in Poland, it is practiced to provide the result of cytological examination in the Bethesda 2001 system and below the Papanicolaou test. Despite the fact that it is possible to compare both scales and match counterparts, we decided to keep the original Papanicolaou cytological smear assessments, also due to the fact that it is often an intuitive scale for Polish medical doctors and patients.
The table summarizing the number of patients in studied groups and the comparison between Pap and Bethesda classification was added to the supplementary materials (Table S3).
It is not clear to me what system of classification of cytological abnormality was used. I,II or III could referral's to the original classification of Papanicolaou of 1963, or one of the Munich systems (see https://www.ncbi.nlm.nih.gov/pmc/articles/PMC4664210/#b2-jtgga-16-4-203). The organized system in Poland is stated to use a modification of the Bethesda system (PMC4417537, as already cited).
We described the classification in Table S3 (Supplementary Materials).
It is stated that the cells obtained by LBC were divided into two parts (one for Raman imaging, the other for HPV testing). How was this done? Could the method of division affect the relationship between imaging and HPV? LBC cytology does not necessarily map to results of histological examination of tissue samples.
The obtained cervical samples were centrifuged and pipetted into two tubes – one of them was used for isolation of genetic material and the other for imaging. The method of division could not anyhow affect the relationship between imaging and HPV.
Referring back to point 1, had any or indeed all of these women undergone colposcopy with large loop excision of the transformation zone (LLETZ, also known as LEEP) or punch biopsy, or other intervention. Such procedures could affect presence of HPV infection and cytology. Indeed, for a mechanistic study, why use cytology rather than histopathological samples?
As mentioned, the studied patients did not undergo any therapeutic interventions or biopsies before sampling. This information was added to the text in the “Clinical specimens” section.
Reviewer 2 Report
ID IJMS-642976
"Impact of HPV infection on the cellular glycogen level: a new molecular mechanism in progression of cervical carcinogenesis. Raman microscopic study with subcellular resolution" to be published in International Journal of Molecular Sciences.
General remarks
The authors presented a study which topic was using Raman microscopy in the investigation of epithelial cervical cells which had been collected from 87 females with squamous cell carcinoma or belonging to groups I,II or III according to Papanicolau test. All females were tested for HPV infection using PCR. Subcellular resolution of Raman microscopy enabled to understand phenotypic differences in a heterogeneous population of cervical cells in the following groups: Pap I/HPV–, Pap II/HPV–, Pap II/HPV+, Pap III/HPV–, Pap III/HPV+ and cancer cells (HPV+). Authors showed for the first time that glycogen content significantly changed with the nucleus size of cervical cells in Pap II and III groups, but not Pap I and cancer groups. For large-nucleus cells HPV infection resulted in additional glycogen depletion compared to HPV negative cells in Pap II and III groups. They hypothesized that accelerated glycogenolysis in large-nucleus Pap II and III cells may be associated with their pro-oncogenic character and increased protein metabolism, particularly for HPV positive cells. Their work underlines unique capabilities of Raman microscopy in single cell studies and demonstrated a potential of Raman-based methods in HPV diagnostics.
The topic of the article is very interesting and it could serve as an important basis for further research.
There are only some minor corrections needed before the article:
line 19: "collcted" -- please correct to "collected"
line 20: "Papanicolau" -- please correct to "Papanicolaou"
line 47: "CIN3 cervical cancer" -- unclear; is it CIN3 or cancer or both?
line 67: remove the comma after "particularly"
line 68: add a comma after "microscopy"
line 76: "moreover, the" -- should be "moreover with the"
line 325: "to discriminated" -- should be "to discriminate"
lines 339-340: the last sentence should be rearranged (e.g., "To visualize nPCR effects, an agarose gel electrophoresis with addition of bromodeoxyuridine (BrDU) was performed.")
line 342: please correct "PAS staining"
line 345: "Cells was" -- please correct to "Cells were"
line 364: the symbol between Pap II/HPV and SCC/HPV is unclear; is it a dash?
line 371: please correct "ovarian cancer cells"
line 425: in reference 10, the list of authors is interrupted by a number ("13.") Are those supposed to be different references?
Author Response
The topic of the article is very interesting and it could serve as an important basis for further research.
We would like to thank the reviewer for evaluating our paper and her/his nice comments.
All proposed corrections were introduced to the text of the manuscript. They are listed below.
There are only some minor corrections needed before the article:
line 19: "collcted" -- please correct to "collected"
It was corrected.
line 20: "Papanicolau" -- please correct to "Papanicolaou"
It was corrected.
line 47: "CIN3 cervical cancer" -- unclear; is it CIN3 or cancer or both?
It was corrected for CIN3 according to the publication.
line 67: remove the comma after "particularly"
It was removed.
line 68: add a comma after "microscopy"
It was added.
line 76: "moreover, the" -- should be "moreover with the"
It was corrected.
line 325: "to discriminated" -- should be "to discriminate"
It was corrected.
lines 339-340: the last sentence should be rearranged (e.g., "To visualize nPCR effects, an agarose gel electrophoresis with addition of bromodeoxyuridine (BrDU) was performed.")
It was rearranged.
line 342: please correct "PAS staining"
It was corrected.
line 345: "Cells was" -- please correct to "Cells were"
It was corrected.
line 364: the symbol between Pap II/HPV and SCC/HPV is unclear; is it a dash?
The symbol means "approximately equal", it was bolded.
line 371: please correct "ovarian cancer cells"
It was corrected.
line 425: in reference 10, the list of authors is interrupted by a number ("13.") Are those supposed to be different references?
It was corrected, it was one reference, "13" was introduced by mistake.
Reviewer 3 Report
This is a very interesting Paper that I enjoyed reading. However there are a few issues that need to be addressed
General comments
I would prefer if the paper was formatted in the general style of introduction, materials and methods, results/discussion and conclusion. Instead of having the methods in the back above the conclusion.
Please highlight the subsections in the results section more clearly.
The result section should have a summary figure of all the results to make it easier to follow
Remove the term FEMALE from the paper, as it is assumed when using cervical samples your sample base is female
I was not familiar with this type of Pap grading (Pap II) ect. A figure should be added which converts it into the more commonly used Bethesda system.
e.g Pap group III = LSIL/HSIL
A section should be added about the HPV genome and how it replicates. For example HPV is a 5,000 base pair virus with 5 early genes E1-E5 and 2 late gene E6, E7 ect. How does it infect ect? Give a brief life cycle of the virus
You need to discuss HPV testing DNA/ mRNA based briefly in the introduction.
What substrate did you use for your cells while recording on Raman?
What cell types are you recording from? Intermediate, superficial, parabasal, basal???
How are you incorporating for the different cell types enlarged nuclei for example. A LSIL sample would only show enlarged intermediate/ superficial cells. HSIL would show both enlarged nuclei in intermediate/superficial and parabasal cells?
How many samples were used in each category?
Specific Comments
Line 40: Remove infection from HPV Infection test.
Line 42: remove swab and replace with smear.
Line 43-45 Remove text and replace with more update statistics from more recent papers (not 2003)
Line 45: Remove conventional smear data as it is no longer relevant
Line 47: Remove CIN 3 cervical cancer. CIN 3 is not cancer. It is a histological grading of pre-cancer.
Line 49: Give the percentage
Line 54: remove term cured. Rewrite sentence to incorporate the fact that cervical abnormalities can regress and HPV can be eliminated from the body.
Line 57: This sentence should be , rewritten or remove. suggestion 'The nucleur to cytoplasm ratio is used as a cytological feature to identify abnormal cells'
Line 63-70. Remove up until Raman spectroscopy
Line 100 ectocervical cells in endocervical cells?? review
Line 180: You mention pap III/HPV negative. If HPV is present in 99% of all cervical cases. Do you think that this is a true (correct classification of this sample). Could it be a false positive? could be Inflammation. Should it be included?
Figure 1 What cell types are you looking at?
Line 192: Check grammer ( as without abnormalities) Suggestion: Patients classified as Pap II usually present as both cytologically and histologically negative.
Line 222-233: It would help to present this data in a table format.
Line 270-281: It would help to present this data in a table format
Line 337: Outline which types of HPV you are testing for? High or low risk Which subtypes? 16,18, 21.45 ?
Author Response
This is a very interesting Paper that I enjoyed reading. However there are a few issues that need to be addressed
We thank the reviewer for very careful reading our paper and detailed comments. Our work was considerably improved due to the reviewer's comments.
The changes in the text were made and tables in the supplementary material were introduced. Below are our answers for comments.
General comments
I would prefer if the paper was formatted in the general style of introduction, materials and methods, results/discussion and conclusion. Instead of having the methods in the back above the conclusion.
Unfortunately, the general format of the manuscript is according to the journal guidelines, therefore we need to preserve it.
Please highlight the subsections in the results section more clearly.
The subsections were highlighted.
The result section should have a summary figure of all the results to make it easier to follow
This is a very good idea. We have introduced such figure (in the new version of the manuscript – Fig. 4).
Remove the term FEMALE from the paper, as it is assumed when using cervical samples your sample base is female.
The term "female" was excluded in the cases where it was redundant, where the noun was necessary, the term "patient" was used.
I was not familiar with this type of Pap grading (Pap II) ect. A figure should be added which converts it into the more commonly used Bethesda system.
e.g Pap group III = LSIL/HSIL
It is a good remark. We added a table in the supplementary material (Table S3) that show the number of patients in each category and conversion to the Bethesda system. In fact, in the Pap III group there were both LSIL and HSIL samples.
A section should be added about the HPV genome and how it replicates. For example HPV is a 5,000 base pair virus with 5 early genes E1-E5 and 2 late gene E6, E7 ect. How does it infect ect? Give a brief life cycle of the virus
The section was added in the introduction.
You need to discuss HPV testing DNA/ mRNA based briefly in the introduction.
The section was added in the introduction.
What substrate did you use for your cells while recording on Raman?
CaF2 slides were used, we added this information in the "Clinical specimens" section.
What cell types are you recording from? Intermediate, superficial, parabasal, basal???
How are you incorporating for the different cell types enlarged nuclei for example. A LSIL sample would only show enlarged intermediate/ superficial cells. HSIL would show both enlarged nuclei in intermediate/superficial and parabasal cells?
It is an interesting remark, however using Raman microscopy we are not able to recognize if a cell is intermediate/superficial or parabasal, therefore cells were recorded randomly for each sample, 10 cells per sample, and a diameter of 10 µm is the only criterion that is used to divide cells into small-nuclei and large-nuclei groups.
How many samples were used in each category?
As I mentioned above, we added a table in the supplementary material (Table S3) that show the number of patients in each category and conversion to the Bethesda system.
Tabelka do uzupełnienia/weryfikacji.
Specific Comments
Line 40: Remove infection from HPV Infection test.
It was removed.
Line 42: remove swab and replace with smear.
It was replaced.
Line 43-45 Remove text and replace with more update statistics from more recent papers (not 2003)
The more recent paper was added to replace the study from 2003.
Line 45: Remove conventional smear data as it is no longer relevant
The conventional smear data were removed.
Line 47: Remove CIN 3 cervical cancer. CIN 3 is not cancer. It is a histological grading of pre-cancer.
It was corrected.
Line 49: Give the percentage
It is very difficult to give one value here, as it is the risk of progression significantly depends of the grade of dysplasia. Therefore we added the information about the percent of patients for whom HPV-induced cervical cell changes regress spontaneously.
Line 54: remove term cured. Rewrite sentence to incorporate the fact that cervical abnormalities can regress and HPV can be eliminated from the body.
It was corrected according to the reviewer's comment.
Line 57: This sentence should be , rewritten or remove. suggestion 'The nucleur to cytoplasm ratio is used as a cytological feature to identify abnormal cells'
It was corrected according to the reviewer's comment.
Line 63-70. Remove up until Raman spectroscopy
The fragment was moved up.
Line 100 ectocervical cells in endocervical cells?? Review
It was corrected for: ectocervical cells, while in endocervical cells.
Line 180: You mention pap III/HPV negative. If HPV is present in 99% of all cervical cases. Do you think that this is a true (correct classification of this sample). Could it be a false positive? could be Inflammation. Should it be included?
Samples were classified HPV -/+ according to PCR reactions. Pap III group in general is not very homogenous, but Pap III samples are not cervical cancer. We do not have reasons to think that our PCR reactions give wrong results, therefore we believe that these are indeed Pap III/HPV negative samples. According to a recent Human Papillomavirus and Related Diseases Report [https://hpvcentre.net/statistics/reports/POL.pdf], in Poland, ca. 60% of HSIL HPV 16 and 18 samples are HPV+ and as much as 40% HPV-.
Figure 1 What cell types are you looking at?
We presented random cervical epithelial cells obtained from random patients belonging to Pap II/HPV– group: cell with a nuclei smaller than 10 mm (A panel) or C: Pap II/HPV+ group: cell with a nuclei bigger than 10 mm (C panel).
Line 192: Check grammer ( as without abnormalities) Suggestion: Patients classified as Pap II usually present as both cytologically and histologically negative.
It was corrected according to the reviewer's comment.
Line 222-233: It would help to present this data in a table format.
We added a table in the supplementary material (Table 1S) that showed the data presented in Fig. 3.
Line 270-281: It would help to present this data in a table format
We added a table in the supplementary material that show the data presented in Fig. 4.
Line 337: Outline which types of HPV you are testing for? High or low risk Which subtypes? 16,18, 21.45 ?
The information about HPV types was added to the text in the “PCR reactions” section.
Round 2
Reviewer 1 Report
I appreciate the responses to my earlier comments. However, the provenance of the women from whom samples were obtained is still unclear to me. Were the cytological abnormalities detected in women who accepted the offer of population-based screening or in women who had more complex histories that led them to be referred to the microbiology research centre or oncology department?
It is stated in the response that the studied patients did not undergo any therapeutic interventions or biopsies before sampling, and that this information has been added to text in "Clinical specimens" section:
Author Response
I appreciate the responses to my earlier comments. However, the provenance of the women from whom samples were obtained is still unclear to me. Were the cytological abnormalities detected in women who accepted the offer of population-based screening or in women who had more complex histories that led them to be referred to the microbiology research centre or oncology department?
We are sorry that our previous explanations were not clear. Hopefully, this time we can clarify more accurately the provenance of women.
The reviewer is of course correct that a cervical screening program is organized in Poland by the Ministry of Health and the National Health Fund. In the framework of this program women aged 25-59 are invited to undergo cervical screening every three years. However, many women prefer to undergo cytological screening in private facilities they trust. One of many such facilities is the Centre Microbiological Research that despite the name referring only to microbiological analysis, performs cervical screening tests.
Therefore, the samples described in our article came from two places:
Centre Microbiological Research that is not a tertiary referral center. Women come to this place for prophylactic check ups and a random group of 76 women was chosen for our research. Clinic of Radiotherapy, Maria Sklodowska-Curie Institute – Oncology Center - the samples obtained from the Oncology Center came from women who had their first visit to this referral center. Samples were taken prior to the therapy and patients at this level did not undergo any therapeutic interventions, radiochemotherapy or biopsies before sampling. The samples were protected and stored until the results of colposcopy and histopathological examination were obtained. We included, in our experiments, cells of confirmed squamous cell cervical cancer from 11 randomly chosen women.To clarify these issues we made changes in the first paragraph of the “Clinical specimens” section and in the current version it looks as follows:
In this study, samples from 87 women from the south of Poland and in the age of 19-76 years old were collected from October 2017 to May 2019 (Fig. 6A). Cervical epithelial cells were obtained from The Centre of Microbiological Research and Autovaccines, in memory of Jan Bobr and from Department of Gynaecological Oncology, Maria Sklodowska-Curie Memorial Cancer Centre and Institute of Oncology. Research included women who underwent prophylactic check ups (samples collected in The Centre of Microbiological Research and Autovaccines) and women during their first visit to the referral Oncology Center. Samples were taken prior to therapy and patients at this level did not undergo any therapeutic interventions, radiochemotherapy or biopsies before sampling. In the Oncology Center the samples were protected and stored until the results of colposcopy and histopathological examination were obtained. We included, in our experiments, cells of confirmed squamous cell cervical cancer.
It is stated in the response that the studied patients did not undergo any therapeutic interventions or biopsies before sampling, and that this information has been added to text in "Clinical specimens" section:
I don't see any such addition to the "Clinical specimens" section I still have a concern that there seems to be no histological follow-up of the cytology test results, after sampling.
We apologize the reviewer here for our mistake, we submitted the last but one version of the manuscript in which we did not include the above-mentioned sentence. In a current version, we have included this information in the text (see above).